# A Traffic Parameter Extraction Model Using Small Vehicle Detection and Tracking in Low-Brightness Aerial Images

**Junli Liu [1], Xiaofeng Liu [1,2,*], Qiang Chen [1,2] and Shuyun Niu [3]**

[1] School of Automotive and Transportation, Tianjin University of Technology and Education, Tianjin 300222, China; serafina19971009@163.com (J.L.); qf_chenqiang@163.com (Q.C.)

[2] National & Local Joint Engineering Research Center for Intelligent Vehicle Road Collaboration and Safety Technology, Tianjin 300222, China

[3] ITS Center, Research Institute of Highway Ministry of Transport, Beijing 100088, China; nsy@itsc.cn

* Correspondence: microbreeze@126.com; Tel.: +86-22-88181527

**Abstract:** It is still a challenge to detect small-size vehicles from a drone perspective, particularly under low-brightness conditions. In this context, a YOLOX-IM-DeepSort model was proposed, which improved the object detection performance in low-brightness conditions accurately and efficiently. At the stage of object detection, this model incorporates the data enhancement algorithm as well as an ultra-lightweight subspace attention module, and optimizes the number of detection heads and the loss function. Then, the ablation experiment was conducted and the analysis results showed that the YOLOX-IM model has better mAP than the baseline model YOLOX-s for multi-scale object detection. At the stage of object tracking, the DeepSort object-tracking algorithm is connected to the YOLOX-IM model, which can extract vehicle classification data, vehicle trajectory, and vehicle speed. Then, the VisDrone2021 dataset was adopted to verify the object-detection and tracking performance of the proposed model, and comparison experiment results showed that the average vehicle detection accuracy is 85.00% and the average vehicle tracking accuracy is 71.30% at various brightness levels, both of which are better than those of CenterNet, YOLOv3, FasterR-CNN, and CascadeR-CNN. Next, a field experiment using an in-vehicle global navigation satellite system and a DJI Phantom 4 RTK drone was conducted in Tianjin, China, and 12 control experimental scenarios with different drone flight heights and vehicle speeds were designed to analyze the effect of drone flight altitude on speed extraction accuracy. Finally, the conclusions and discussions were presented.

**Keywords:** traffic information and control; vehicle detection and tracking; unmanned aerial vehicles; low brightness images

## 1. Introduction

Road traffic monitoring is the foundation of intelligent transportation systems (ITS), which can obtain road operation situations, such as vehicle trajectory, vehicle speed, and vehicle type. This is crucial for further alleviating traffic congestion and improving traffic safety. In recent years, unmanned aerial vehicles (UAVs) loaded with high-resolution cameras have been introduced to collect real-time traffic information [1,2]. UAVs have a greater flexibility to record real-time traffic data than fixed detectors (LIDAR, millimeter wave radar, and induction loop detectors, etc.). Moreover, with the development of computer vision and deep-learning neural networks, UAVs have attracted increasing attention in the field of vehicle operation parameter extraction. Considering the limited road coverage, the accuracy of urban road data collection by fixed traffic cameras is only between 50% and 75% [3]. Due to the unique advantages of their wide field of view and mobility, the accuracy of the UAV-based urban road data acquisition system exceeds 80% [4,5].

Vehicle detection is one of the most important steps in achieving traffic monitoring. Appearance-based and motion-based detection models are the two main methods for vehicle detection. Since the size, shape, and color of vehicles are always different, the

appearance-based method is often adopted. For example, Tsai et al. [6] fused three features, such as color, edge, and corner points, to complete vehicle detection. As the number of scenarios for vehicle detection applications continues to increase and the requirements for features continue to improve, local feature descriptors enter the thinking of scholars. Among them, the histogram of oriented gradient (HOG), local binary pattern (LBP), Haar-like, Gabor, and other local feature descriptors are widely used in vehicle detection. Geng et al. [7] improved HOG features and combined them with LBP features to obtain the fused features of vehicles. Wen et al. [8] used a Haar-like feature extraction method to represent the edges and structures of vehicles. Tang et al. [9] used two features, Haar-like and Gabor, to locate the vehicles in pictures, respectively. However, vehicles on the road are generally in motion and, in addition to the appearance information that can be used for vehicle detection, motion-based detection methods have been introduced. Ji et al. [10] proposed the inter-frame difference method with five adjacent frames. Weng et al. [11] introduced the three-frame difference method with background subtraction to obtain the grayscale map, and then the motion objects were effectively extracted by the morphological algorithm. Liu et al. [12] combined the three-frame difference method with an Adaboost classifier trained with Haar-like features to detect the moving vehicles. Moreover, detecting small vehicles in aerial imagery presents a significant challenge. Zhang et al. [13] introduced deformable convolution into the backbone network [14] to help the model learn geometric transformation capabilities efficiently to improve the performance of small object detection through data enhancement and improvements. Tian et al. [15] presented a dual neural network-based approach for detecting small objects. The method employs a secondary detection mechanism to optimize the detection of objects with low confidence and missed detection. However, the detection accuracy of these traditional algorithms is largely affected by the object detection distance as well as the complex background, so the above algorithms are only suitable for detection scenarios with similar target shapes and fixed backgrounds. The aerial image background is wider and there are more types of vehicles, so these traditional algorithms are not suitable for UAV-based traffic monitoring.

In recent years, with the development of deep learning, the innovation of deep neural network models has opened new ideas for object detection, and greatly improved the accuracy of object detection. Among them, the two-stage object detection networks R-CNN [16], fast-RCNN [17], Faster-RCNN [18], etc. and the one-stage detection networks YOLO, SSD, etc. have achieved great victories in object detection. R-CNN, as a pioneer of two-stage detection, uses a selective search algorithm to propose a series of regions of interest (RoI) that may contain a target at the first stage. In addition, at the second stage, CNN is used to extract deep features within the regions, classify, and localize them accordingly. Fast-RCNN puts the position of the convolutional layer in front, and transforms the feature extraction process of each ROI into the feature vector through the position mapping relationship between images and features, which reduces the calculation amount and computation time at the second stage. Fast-RCNN and Faster-RCNN shorten the computation time of the first and second stages respectively based on R-CNN, but the computation for a large number of redundant regions will inevitably pull down the detection efficiency, so these algorithms do not perform well during real-time traffic monitoring scenarios.

The YOLO algorithm [19] is a representative work of one-stage detection algorithms. It first divides the image into grids of equal size but varying resolutions, and places anchor frames of different sizes and shapes at their vertices. These anchor frames have a position mapping relationship with the features extracted by the CNN, allowing for the prediction and regression of the features to obtain target classes and positions in the anchor frames. The YOLO algorithm replaces the unquantified and heavily redundant regions in two-stage algorithms with a fixed number of anchor frames, which significantly improves detection efficiency and meets real-time detection. Li et al. [20] proposed a multi-object vehicle detection algorithm based on YOLOv2 for different roadway scenarios. Lin et al. [21] proposed the feature pyramid network (FPN), which effectively improves the detection performance of small objects by fusing high-level semantic information with low-level

semantic information. Van et al. [22] proposed a method called YOLT for multi-scale object detection in satellite images by modifying YOLOv2. Yang et al. [23] proposed SCRDet to effectively capture small objects in complex backgrounds in UAV aerial images as well as to address the diversity of object directions. Rajput et al. [24] proposed an improved YOLOv3-based multi-vehicle detection algorithm for multi-vehicle detection in complex traffic scenarios such as parking lots, urban roads, and highways. Zhang et al. [25] proposed an improved method of vehicle detection in different traffic scenarios based on an improved YOLO v5 network. The proposed method uses the Flip-Mosaic algorithm to enhance the network's perception of small targets. These studies have validated the effectiveness of the YOLO series models for object detection in traffic scenarios, but there is no comprehensive consideration of how to balance the accuracy and efficiency of object detection, nor is there any discussion of the special conditions for traffic scenarios.

In summary, different strategies have been adopted to improve vehicle detection accuracy, however, it is still a challenge to balance the object detection efficiency and accuracy. In this condition, lightweight models are worth further studying. In addition, the above studies mainly conduct vehicle detection and tracking under high-brightness conditions; few low-brightness experiments have been implemented to verify the model performance. Moreover, with the UAV height increasing, the size of the detected target will decrease, and the effect of UAV heights on detection accuracy is seldom considered.

The contributions of this study are listed as follows:

(1) Data enhancement. This paper extends the low-brightness aerial image dataset by using data enhancement strategies, such as the slicing-aided hyper inference (SAHI) algorithm, coordinate correction matrix, HSV perturbation, and brightness enhancement, which enhance the robustness and generalization ability of the model in detecting small targets with complex backgrounds.

(2) Model optimization. In this paper, an improved YOLOX-IM vehicle detection algorithm is proposed. In order to balance the detection efficiency and accuracy, the model incorporates the ultra-lightweight subspace attention module (ULSAM) in the path aggregation network (PAN). In addition, in order to achieve a lightweight model, the boundary regression loss function is optimized, and the SIoU loss function is used to optimize the model, making the boundary regression faster and more accurate.

(3) Field experiment verification. The YOLOX-IM object detection model is connected with the DeepSort target tracking model, and then a vehicle speed estimation algorithm is fused to construct a UAV-based traffic parameter extraction model. In this study, experimental vehicles equipped with global navigation satellite systems (GNSS) and on-board diagnostics (OBD) in particular are used, to collect real traffic parameters to verify the accuracy and applicability of the proposed model. In addition, the effect of UAV flight altitudes on traffic parameter extraction accuracy is analyzed.

The rest of this paper is structured as follows: Section 2 introduces the YOLOX-s model and the processing strategy for the low-brightness aerial image dataset. In Section 3, the YOLOX-IM-DeepSort multi-object detection and tracking model is proposed, and the improvements are specified. Section 4 provides a description of the experimental design and conducts the experiment verification result. Finally, in Sections 5 and 6, conclusions and discussions are given, respectively.

## 2. Baseline Model and Dataset Processing

### 2.1. YOLOX-s Model

The YOLOX object detection framework was proposed in 2021 [26], and it is a competitive model in object detection competitions. Small objects are difficult to detect due to their low resolution and small size. The algorithm architecture of YOLOX-s, on the other hand, improves the number of unanchored points and the Intersection-over-Union (IOU), which makes the feature map more informative. Not only that, the combination of downsampling and upsampling used in this architecture improves the sampling resolution of the feature map, which ultimately leads to the further enhancement of the detection of

small objects. In order to improve the accuracy and speed of object detection, this paper adopts the YOLOX-s model as the baseline model.

Cross stage partial darknet (CSPDarknet) is the backbone feature extraction network of YOLOX-s, which consists of five parts: DARK1, DARK 2, DARK 3, DARK 4, and DARK 5. The backbone network structure is shown in Figure 1; the size of the feature map is halved by downsampling between each section, while the number of channels between each section is doubled. CSPDarknet uses the focus module [27], residual network [28], cross stage partial network (CSPnet) [29], and spatial pyramid pool (SPP) [30] to extract image features. Among them, the focus module reduces the number of added parameters and increases the operational efficiency without losing information. The residual network deals with the problem of gradient disappearance by increasing the depth of the neural network. CSPnet enhances the ability of the network to learn features while reducing calculating workload. The SPP module uses the maximum pooling operation of different pooling kernels to improve the perceptual field of the network. The object detection process of YOLOX-S is as follows. Firstly, the images are input to the backbone network for feature extraction through CSPDarknet. Then, the effective feature maps are passed to the neck structure, and the features are enhanced by PAN to obtain the three-layer enhanced feature maps. Here, the feature pyramid networks (FPNs) propagate the top-down high-level semantic information into the shallow feature maps, enhancing the semantic information at multiple scales, but with weak localization information of the objects. Therefore, a bottom-up path is added to PAN, which transfers rich spatial information from shallow to higher-level feature maps and enhances target localization on multiple scales. Finally, the decoupled head classifies and regresses the three enhanced feature maps, which improves the convergence speed and the detection accuracy.

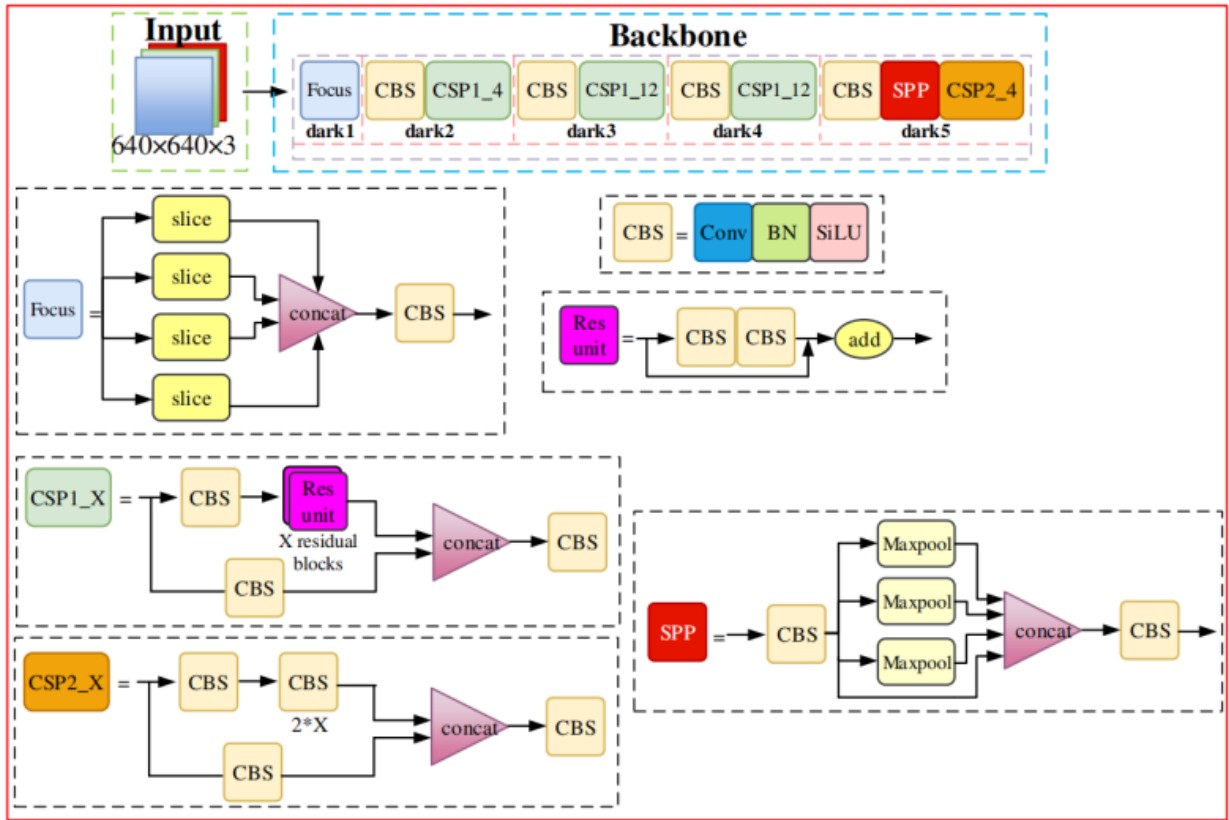

**Figure 1.** Backbone network structure of YOLOX-s.

### 2.2. Processing Strategy for Aerial Image Dataset

Real-life aerial images include daytime and nighttime images, as well as aerial images of special weather conditions, such as rain, haze, cloudy skies, etc. The data volume and distribution will affect the training direction and result of the YOLOX-s model, so it is necessary to select the appropriate dataset according to the application scenarios. The Highway Drone Dataset [31] and the Visdrone Dataset [32] have been used to train and test the object detection model in aerial images, which are multi-angle aerial images, and therefore are more suitable for traffic object detection and monitoring than other common available datasets. The impact of different brightnesses on object detection is significant; only using the base dataset for model training would ignore the effect of low-brightness on object detection. Data augmentation is a common strategy for training YOLOX-S models to help extend the data diversity, enhance the generalization of the network, and avoid training the network to local loss minima [33]. HSV domain perturbation [34] randomly changes the hue, saturation, contrast, and luminance of the image with 50% probability during training, expanding the color distribution of the images in the dataset, and making the model enhance the generalization to image color and luminance changes, as shown in Figure 2.

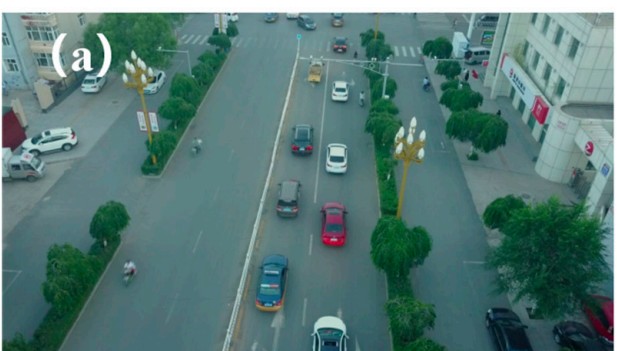 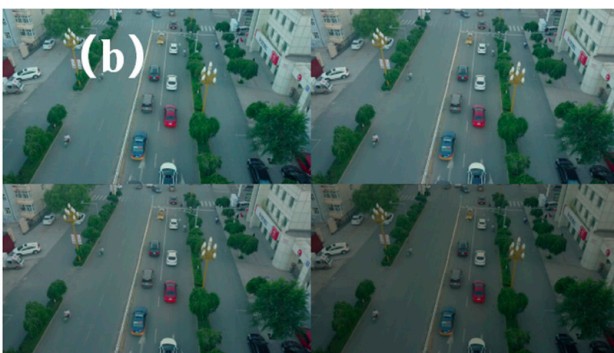

original image         data-augmented image

**Figure 2.** HSV random perturbation.

By random perturbation in the HSV domain, multiple changes in the color and brightness of each image are acquired. Although the color and light distribution of the dataset is expanded, the perturbed images still have a large gap with the aerial images at night. For this reason, this study analyzes the gap between the brightness of daytime and nighttime images, and compensates for the gap with the actual detection by adjusting the image brightness in training, as described in Section 3.

### 3. Traffic Parameter Extraction Methods and Material

The traffic parameter extraction process is described in Figure 3. The model will first pre-process the UAV images under different schemes, then output to the optimization model for object detection and tracking and, finally, extract the traffic operation parameters of the target vehicles.

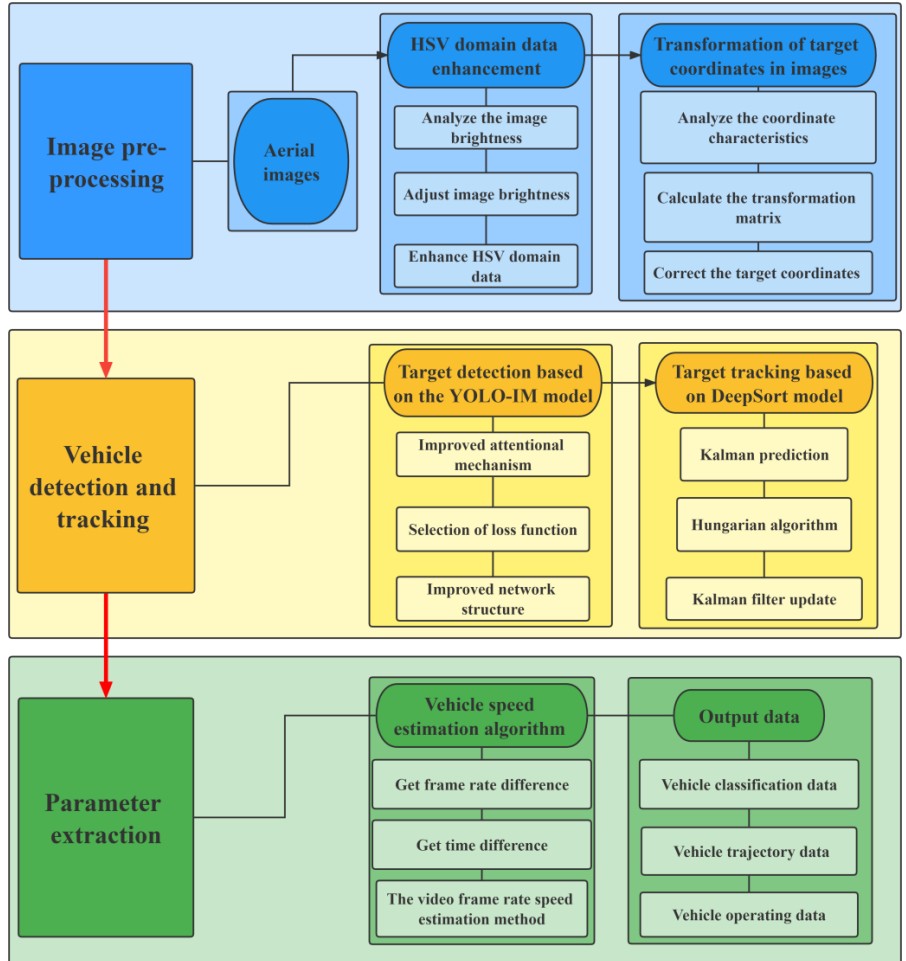

**Figure 3.** Flow chart of aerial video traffic parameter extraction.

### *3.1. Image Pre-Processing*

### 3.1.1. HSV Domain Data Enhancement

The most obvious difference between the aerial images at night and daytime is illumination. Comparing the brightness of the training set and the test set images in the HSV domain in the Visdrone2021 dataset, the mean value of brightness of all the daytime images in the test set is 112.86, which is equal to the training set, while the mean value of brightness of the night images is 63.66. A normalized histogram of the mean image brightness in the test set was compared, as shown in Figure 4.

As shown in Figure 4, the difference between the mean night image brightness and the mean day image brightness is greatest at 93.75, that is, only 8.05% of the total daytime images have a mean brightness that is less than 93.75, and 10.38% of the total nighttime images have a mean brightness greater than 93.75. Adjusting the brightness of all images to 93.75 during training not only simulates the brightness of dusk, but also enhances the generalization of object detection in the model at low brightness. Specifically, the brightness processing is performed to judge the illumination conditions of the image through the mean brightness before inference, and to adjust its brightness to 93.75 to simulate the image into a dusk scene. Since adjusting brightness in the RGB color gamut requires calculating the image brightness before adjusting the values of R, G, and B, the brightness value can be directly adjusted in the HSV domain. In order to reduce the loss of accuracy in the calculation process and the additional time needed for brightness calculation, YOLOX-IM incorporates brightness processing into the HSV domain and performs it together with HSV disturbances, collectively referred to as HSV domain data enhancement.

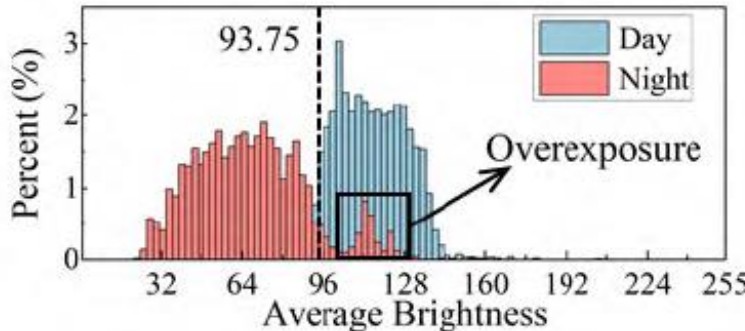

**Figure 4.** Normalized histogram of the image's mean brightness in Visdrone2021 dataset.

In addition to reducing the luminance gap between daytime and nighttime images, HSV domain data enhancement also weakens the overexposure caused by excessive illumination from lights. Figure 5 shows that the brightness average of some nighttime images is between 105 and 127.5, which is caused by the large amount of vehicle lights and road illumination at night. The images taken during the daytime have fewer lights visible and less or no overexposure. Therefore, the features learned by the neural network after training in a training set containing only daytime images tend to portray complete vehicles or pedestrians; the brightness of vehicle lights in images taken at night differs greatly from their surroundings, and the shape of the vehicles is corrupted by the lights and halo effects, which will make it difficult for the neural network to fit vehicles at night and make the model poorly generalized. The HSV domain brightness enhancement strategy proposed in this paper unifies the mean brightness value of images to 93.75, so that the brighter images at night have lower brightness and the darker images during the day have higher brightness, thus reducing the interference of brightness on object detection. The specific effect of HSV domain data enhancement is shown in Figure 5.

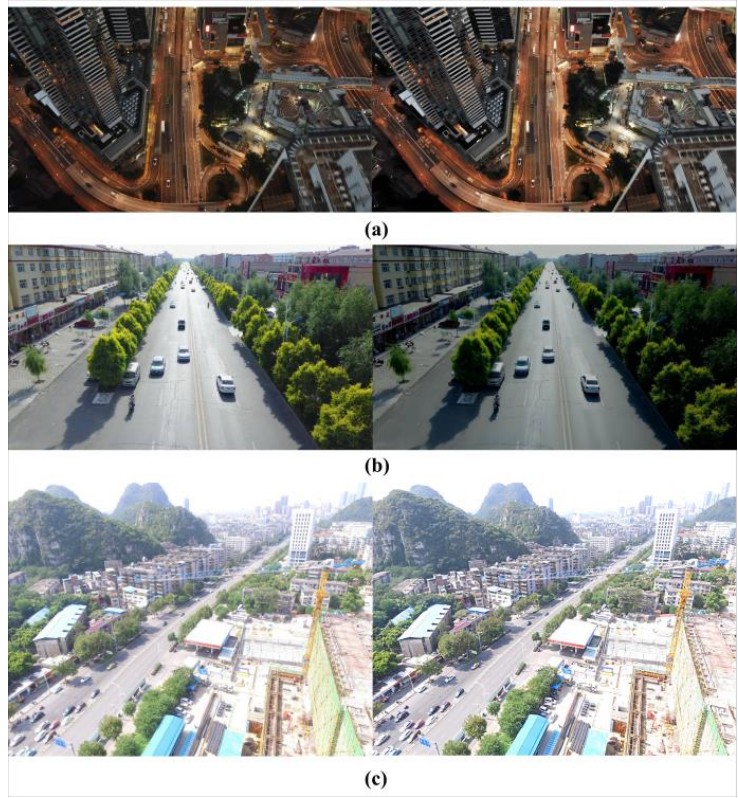

**Figure 5.** Comparison of HSV domain data enhancement effects (Group (**a**) is to increase the brightness, Group (**b**) is to decrease the brightness, and Group (**c**) is to suppress the overexposure).

### 3.1.2. Transformation of Target Coordinates in Images

Wind and UAV attitude can lead to coordinate changes and image ghosting of the target (or road marker points) in UAV aerial video. Therefore, coordinate transformation for target pre-processing in aerial video is required [35].

(1) Coordinate transformation: This paper performs the necessary transformation of real-world coordinates and coordinates of video images by using artificial marker points as a medium. Coordinate transformation can be achieved by establishing an equation relationship between the marker coordinates in the real world and the artificial marker coordinates in the image [36], and the coordinate change model is denoted in Equation (1):

$$\begin{pmatrix} u \\ v \\ p \end{pmatrix} = T * \begin{pmatrix} x_0 \\ y_0 \\ z_0 \end{pmatrix} = \begin{pmatrix} A & D & G \\ B & E & H \\ C & F & I \end{pmatrix} * \begin{pmatrix} x_0 \\ y_0 \\ z_0 \end{pmatrix} \tag{1}$$

In Equation (1), $\begin{bmatrix} u & v & p \end{bmatrix}$ are the coordinates of the target in the real world. $\begin{bmatrix} x_0 & y_0 & z_0 \end{bmatrix}$ are the coordinates of the target in the video made by using the reference system. $T$ is the transformation matrix between real-world coordinates and video image coordinates. In the experiments of this paper, the maximum height of the UAV is less than 120 m, the selected road length is less than or equal to 150 m, and the ratio of road length and flight height of the UAV is much larger than the slope difference, so the spatial three-dimensional problem can be converted into a planar transformation calculation, that is, $C$, $F$, and $I$ in matrix $T$ are equal to 0, 0, and 1 [37], respectively. Therefore, Equation (1) can be written as Equation (2):

$$\begin{pmatrix} u \\ v \\ 1 \end{pmatrix} = T * \begin{pmatrix} x_0 \\ y_0 \\ 1 \end{pmatrix} = \begin{pmatrix} A & D & G \\ B & E & H \\ 0 & 0 & 1 \end{pmatrix} * \begin{pmatrix} x_0 \\ y_0 \\ 1 \end{pmatrix} \tag{2}$$

where $A$, $B$, $D$, $E$, $G$, and $H$ are the parameters to be determined. At least three corresponding marker points are needed to determine the matrix. The more marker points there are, the more robust the transformation matrix will be. Ideally, if there are enough marks on the road, the transformation matrix equation between real coordinates and video image coordinates can be constructed through the matrix obtained by Equation (2).

(2) Image correction: Due to the change in wind force or the adaptive attitude mode of the UAV, the UAV will inevitably move slightly and shake when hovering. Therefore, the transformation relationship needs to be determined to unify the time-varying coordinates between different video frames and define it as an image correction. Equation (3) can be used to represent the transformation process:

$$\begin{pmatrix} x_n \\ y_n \\ 1 \end{pmatrix} = W * \begin{pmatrix} x_0 \\ y_0 \\ 1 \end{pmatrix} = \begin{pmatrix} a_1 & a_2 & a_3 \\ a_4 & a_5 & a_6 \\ 0 & 0 & 1 \end{pmatrix} * \begin{pmatrix} x_0 \\ y_0 \\ 1 \end{pmatrix} \tag{3}$$

where $\begin{bmatrix} x_0 & y_0 \end{bmatrix}$ is the coordinate of the reference coordinate system $N_0$ (initial point) and $\begin{bmatrix} x_n & y_n \end{bmatrix}$ is the coordinate of the $n$th coordinate system. $W$ is the transformation matrix, which consists of 6 parameters. $a_1$ to $a_6$ need to be determined by the fixed markers in the different images. Therefore, the aerial video images need to be corrected by the above two transformation matrices to improve the accuracy of subsequent object detection.

### 3.1.3. Small Target Data Enhancement Strategy

Traditional detection methods are not effective in detecting small objects, and Akyon et al. [38] proposed the Slicing-Aided Hyper Inference (SAHI) method to improve the ability to detect small objects by segmenting the image into multiple overlapping slices so that small objects occupy more pixels in these slices. This paper uses SAHI to crop each image in the training set into 160 sliced images with $640 \times 640$ overlapping pixels, and then removes the sliced images that only contain the background.

Data augmentation can increase the number and diversity of training samples and improve the robustness of the model. Photometric distortion and geometric distortion are the two most commonly used data enhancement methods. Photometric distortion adjusts the brightness, chromaticity, contrast, and saturation of an image. To perform geometric distortion, images can be randomly scaled, cropped, flipped, rotated, or otherwise transformed.

There are other data enhancement methods, such as Cutout [39], Mixup [40], Cut-Mix [41], and Mosaic [42]. Cutout randomly crops a square region of the training image and cuts out the image with zero fill. Mixup randomly selects two samples and their corresponding labels in the dataset and aggregates them using a certain ratio to generate new samples and labels. CutMix randomly selects two images from the dataset and then superimposes the cropped part of one image onto the other. Mosaic randomly crops four images and then stitches them together into a new image that enriches the background of the image. This study uses the photometric distortion, geometric distortion, Mixup algorithm and Mosaic algorithm for small target data enhancement.

### 3.2. Improvement in Object Detection Model

#### 3.2.1. Improved Attentional Mechanism

The attention mechanism operates similarly to the human visual attention mechanism by extracting key features from a larger set of features. By selectively focusing on important features and reducing or ignoring unimportant features, the attention mechanism can enhance network performance. Although the existing attention mechanism can effectively improve the performance of the network, it increases the arithmetic power requirement and is not applicable to lightweight networks. To this end, Saini et al. [43] proposed ULSAM, which can efficiently learn cross-channel information in feature subspaces with a reduced number of parameters. The structure of ULSAM is shown in Figure 6. The input feature mappings are partitioned into *G* mutually exclusive subspaces on the channel, each with *G* feature mappings, and a different attention mapping is derived for each subspace.

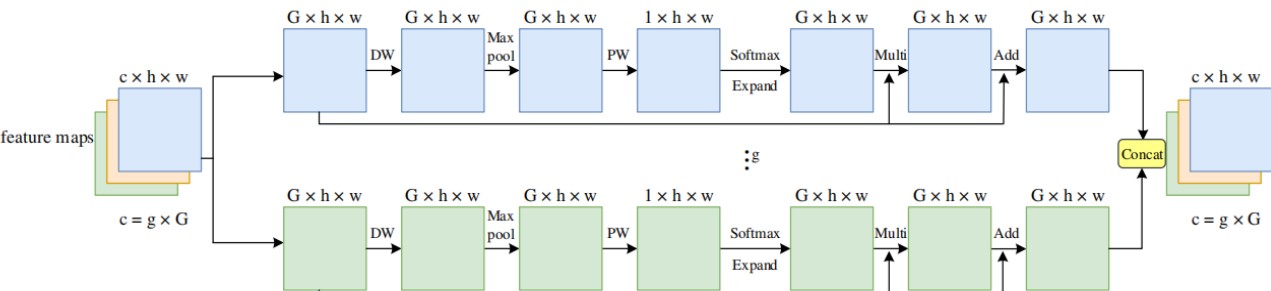

**Figure 6.** Structure of ULSAM.

In this paper, ULSAM is added to the PAN structure for highlighting the object features and weakening the background features. Then, the enhanced object features are input into the decoupled head for classification and regression, which effectively improves the detection accuracy. The improved neck structure is shown in Figure 7.

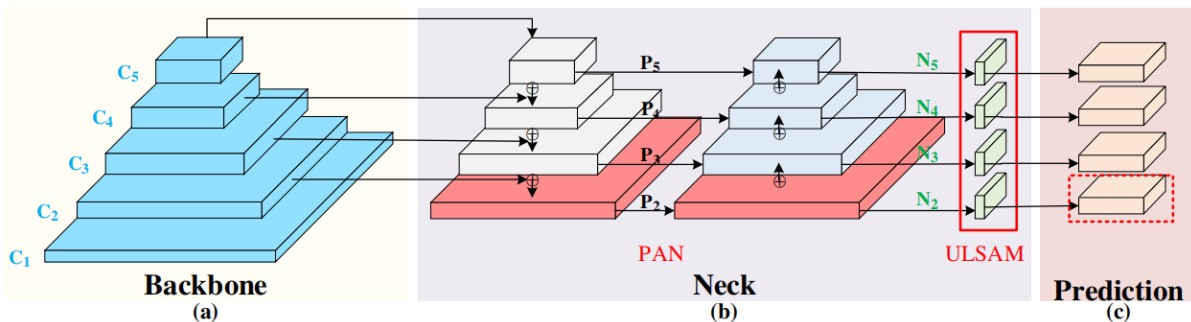

**Figure 7.** YOLOX-IM backbone network architecture.

### 3.2.2. Loss Function Selection

YOLOX uses the IoU loss [44] as the boundary regression loss function. The IoU loss is defined as follows:

$$L_{IoU} = 1 - \frac{\left| b \cap b^{gt} \right|}{\left| b \cup b^{gt} \right|} \tag{4}$$

Here, $b$ indicates the bounding box, and $b^{gt}$ indicates the true value. However, the IoU loss has two drawbacks: firstly, when the bounding box does not intersect the true value, $L_{IoU}$ is a constant value of 1 and the gradient is 0, which can lead to parameters that cannot be optimized. Secondly, when the intersection of the bounding box and the true value is fixed, IoU cannot reflect the way the two boxes intersect. Therefore, this paper replaces SIoU [45] loss with IoU loss to improve the training speed and prediction accuracy of the network. SIoU introduces the vector angle between the true value and the bounding box and redefines the loss function, which consists of four components: angle loss, distance loss, shape loss, and IoU loss. The angular loss is defined as follows:

$$\Lambda = 1 - 2 * sin^2 \left( arcsin(\frac{c_h}{\sigma}) - \frac{\pi}{4} \right) \tag{5}$$

$$\sigma = \sqrt{\left( b_{c_x}^{gt} - b_{c_x} \right)^2 + \left( b_{c_y}^{gt} - b_{c_y} \right)^2} \tag{6}$$

$$c_h = max(b_{c_y}^{gt}, b_{c_y}) - min(b_{c_y}^{gt}, b_{c_y}) \tag{7}$$

Here, $\sigma$ is the distance between the true value and the boundary centroid, $c_h$ is the height difference between the true value and the boundary centroid, $b_{c_x}^{gt}$, $b_{c_y}^{gt}$ are the center coordinates of the ground truth, and $b_{c_x}$, $b_{c_y}$ are the center coordinates of the bounding box. The distance loss is defined as follows, where $c_w$ and $c_h$ are the minimum outer rectangle of the ground truth and the width and height of the bounding box, respectively:

$$\Delta = \sum\nolimits_{t=x,y} \left( 1 - e^{-\gamma \rho t} \right) \tag{8}$$

$$\rho_x = \left( \frac{b_{c_x}^{gt} - b_{c_x}}{c_w} \right)^2 , \rho_y = \left( \frac{b_{c_y}^{gt} - b_{c_y}}{c_h} \right)^2 , \gamma = 2 - \Lambda \tag{9}$$

The shape loss is defined as follows:

$$\Omega = \sum\nolimits_{t=w,h} \left( 1 - e^{(-wt)} \right)^\theta \tag{10}$$

$$\omega_w = \frac{\left| w - w^{gt} \right|}{max(w, w^{gt})} , \omega_h = \frac{\left| h - h^{gt} \right|}{max(h, h^{gt})} \tag{11}$$

where $(w, h)$ and $(w^{gt}, h^{gt})$ are the width and height of the bounding box and ground truth, respectively, and $\theta$ controls the degree of attention to shape loss with values ranging from 2 to 6.

The IoU loss is defined as follows, where $b$ is the bounding box and $b^{gt}$ is the ground truth:

$$IoU = \frac{\left|b \cap b^{gt}\right|}{\left|b \cup b^{gt}\right|} \tag{12}$$

The final SIoU is stated as follows:

$$L_{SIoU} = 1 - IoU + \frac{(\Delta + \Omega)}{2} \tag{13}$$

SIoU redefines the loss function by introducing the vector angles between regressions, which effectively speeds up the training and further improves the accuracy of inference.

### 3.2.3. Improvement in Network Structure

Since the VisDrone2021 dataset [46] has many small objects, the three decoupled heads of YOLOX-s can lead to a large number of missing and false detections when detecting these objects. In contrast, the shallow feature map has a smaller field of perception and stronger spatial information, which is suitable for detecting small objects. The YOLOX-s backbone network extracts features through CSPDarknet and, after 5 rounds of downsampling to obtain a 5-layer feature map $\{C_1, C_2, C_3, C_4, C_5\}$, the number of channels is doubled while the size of the feature map is reduced by half. The feature map $C_1$ contains more detailed information, but also contains high levels of noise and background information. Therefore, in the proposed method, the low-level high-resolution feature map $C_2$ is introduced into the FPN, and the tertiary effective feature maps $\{C_3, C_4, C_5\}$ are fused with top-down transmitted features, and these methods enhance the semantic representation on multiple scales by transferring strong semantic features from high-level to shallow levels. These enhanced feature maps $\{P_2, P_3, P_4, P_5\}$ transfer the strong spatial features of the shallow feature maps to higher levels in a bottom-up manner, improving the localization capability at multiple scales. In addition, a decoupling head for small object detection is added, and the improved network structure is shown in Figure 7.

As shown in Figure 7, the backbone network structure of the YOLOX-IM model uses CSPDarknet to extract the multiscale feature maps, and uses PAN and ULSAM to enhance the multiscale feature maps at the neck layer, so that the FPN is fused with $C_2$ to obtain $P_2$, and $P_2$ is fused bottom-up to obtain $n_2$; $n_2$ and $P_2$ are indicated by red squares. The ULSAM structure is added after the PAN structure to weaken the complex background information, as shown in the red box. In the prediction layer, the red dashed boxes indicate the addition of the decoupled head dedicated to small object detection.

### 3.3. Target Tracking Algorithm

This study adopts the DeepSort multi-objective tracking algorithm [47], which uses more reliable metrics than the Sort tracking algorithm, rather than correlation metrics, thereby increasing the network robustness regarding missing and false objects. The Deep-Sort algorithm architecture is shown in Figure 8. It is worth noting that the input value of DeepSort is the output value of the YOLOX-Im model above.

The DeepSort algorithm includes the following steps: input the vehicle fit detection set and track set to get bbox, then generate Detections dataset, and track dataset; the second step is the Kalman prediction—according to the vehicle history track set, the vehicle position set in the nth frame can be predicted; use the Hungarian algorithm to match the predicted tracks with the detections in the current frame (cascade matching and IOU matching); the Kalman filter update, which outputs the updated spatio-temporal data of the vehicle, which consist of time-variant data and vehicle coordinate data.

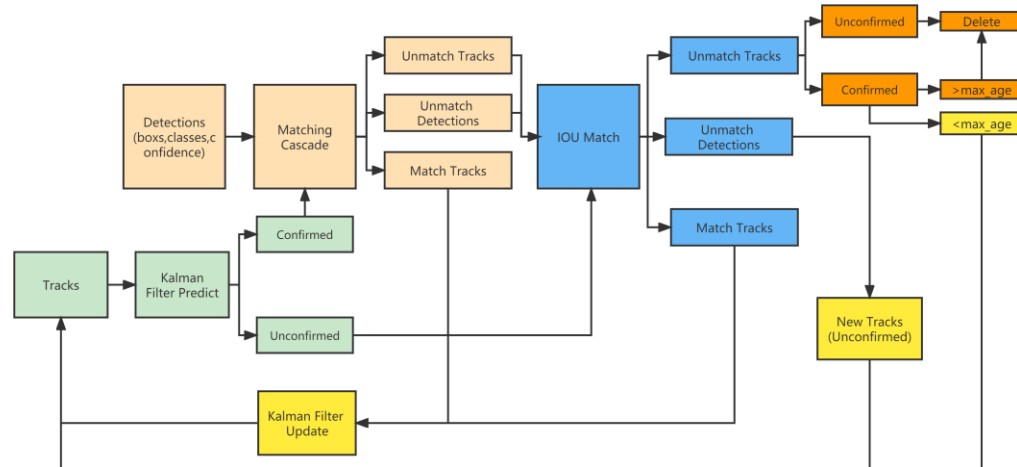

**Figure 8.** DeepSort algorithm architecture.

### 3.4. Output Data

(1) Vehicle classification data. Detect the vehicle objects in the frame using the YOLOX-IM model, and output data as the vehicle fit detection set $N = \{N_i, i = 1,2,3\}$.

(2) Vehicle track data. Combine the dataset output by YOLOX-Im model, use the DeepSort multi-objective tracking algorithm to track the running vehicles in the video and the output data as the vehicle trajectory set $T = \{T_i, i = 1,2,3, \dots \}$.

(3) Vehicle speed data. The video frame rate speed estimation method [48] is used to estimate the speed of the target vehicle. It is important to note that the video frame rate interval is as small as possible in order to reduce the number of calculations and speed up the algorithm execution. In practice, the algorithm takes the speed estimation every 4 frames because of the frequency of in-vehicle data acquisition. The calculation Equation (14) of the video frame rate speed estimation method is listed as follows:

$$v = \frac{\Delta x}{\Delta t} \tag{14}$$

In Equation (14), $v$ is in meters/second, and the vehicle position between the selected frame intervals is subtracted to obtain the moving distance, $\Delta x$, corresponding to the interval, $\Delta t$; $\Delta t$ = (number of frames processed in the previous frame—number of frames processed in the current frame)/frame rate. $\Delta x$ is the distance of the horizontal line of the target vehicle coordinates in the video at different frame rates.

## 4. Experiments and Results

### 4.1. Experimental Design

To verify the performance of the proposed YOLOX-IM model, an ablation experiment is conducted. Then, the YOLOX-IM model is compared with a series of commonly used object detection models using the evaluation metrics. In addition, a practical application is also carried out. The vehicle speed obtained by processing the actual aerial video through the model is denoted as the experimental sample value, and the vehicle speed collected through OBD and GNSS is denoted as the ground sample true value. Then, the error between the experimental sample value and the ground sample true value is compared, which is denoted as the control experiment.

The control experiment mainly consists of an aerial UAV part and the on-board equipment part of the experimental vehicle. For the aerial UAV, a DJI Phantom 4 RTK UAV was used. The GNSS ensures the accuracy of the positioning data and the OBD can record the vehicle driving data in real time by reading the vehicle controller area network (CAN). The experimental system is shown in Figure 9.

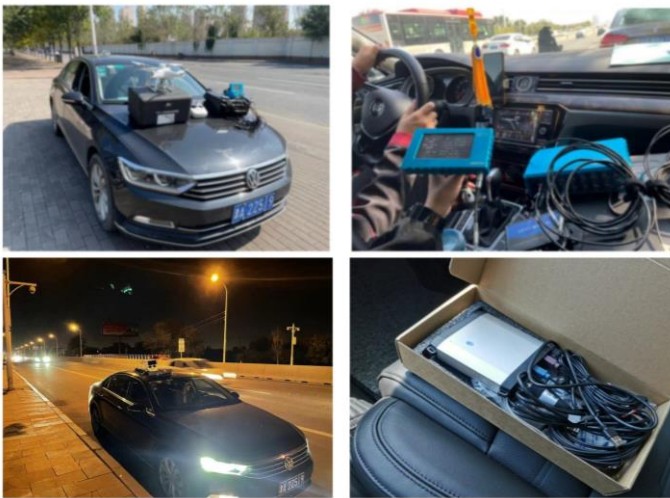

**Figure 9.** Field experimental equipment.

In order to verify whether the YOLOX-IM model can meet the requirements of traffic monitoring under various brightnesses, the control experiment was designed with 12 traffic scenarios each in daytime and nighttime, where the UAV height was set as the three most commonly used monitoring heights, and the speed of the experimental vehicles was also classified, as shown in Table 1. In the control experiments, the UAV hovered at a fixed altitude. The experimental vehicle was driven in a circle along the JinGang Expressway in Tianjin, China, and its trajectories are shown in Figure 10. It is worth noting that since the experimental location of this paper is Tianjin, China, the Chinese letters appearing in Figure 10 only indicate the place name and do not have any other meaning. The line segment in Figure 10 indicates the vehicle trajectory, and the English information above is the vehicle operation data.

**Table 1.** Control experiment with 12 traffic scenarios.

| Experimental No. | Average Recording Time of Drone Aerial Photography (s) | | Speed of the Experimental Vehicle (km/h) | Height of the Drone (m) | Length of Experimental Road (m) |
|---|---|---|---|---|---|
| | **Day** | **Night** | | | |
| 1 | 15 | 18 | 40 | 30 | 50 |
| 2 | 10 | 15 | 60 | 30 | 50 |
| 3 | 23 | 25 | 10~40 | 30 | 50 |
| 4 | 18 | 25 | 10~60 | 30 | 50 |
| 5 | 19 | 27 | 40 | 50 | 75 |
| 6 | 13 | 20 | 60 | 50 | 75 |
| 7 | 27 | 35 | 10~40 | 50 | 75 |
| 8 | 21 | 30 | 10~60 | 50 | 75 |
| 9 | 27 | 30 | 40 | 100 | 150 |
| 10 | 24 | 35 | 60 | 100 | 150 |
| 11 | 33 | 40 | 10~40 | 100 | 150 |
| 12 | 28 | 34 | 10~60 | 100 | 150 |

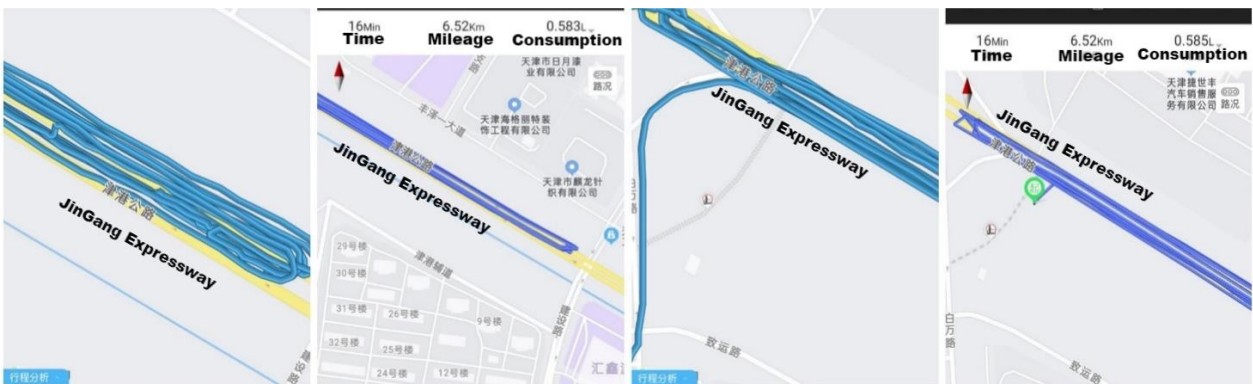

**Figure 10.** The trajectory of the experimental vehicle for the control experiment.

*4.2. Comparison Experiments*

4.2.1. Dataset Description

The VisDrone2021 dataset was collected by Tianjin University, China, and includes a variety of scenes from 14 cities under different weather and lighting conditions. The dataset provides images taken by drones at different heights and locations, with image resolutions of up to 2000 × 1500 pixels. It includes 10 categories: pedestrians, people, bicycles, cars, vans, trucks, tricycles, awning-tricycles, buses, and motors. The training set contains 6471 images, the validation set contains 548 images, and the test set contains 1610 images. In the VisDrone dataset, 902 objects can appear in a single image, and the distribution of categories and labels is shown in Figure 11. To make the category distribution more balanced, this study used SAHI to crop each training image into slices with a 640 × 640 resolution and 160 overlapping pixels, and removed slices that contained only background. Next, selected slices containing fewer sample categories were used to form a new training set with the original training images, 548 images were selected as the validation set, and experimental results were obtained by testing 1610 images.

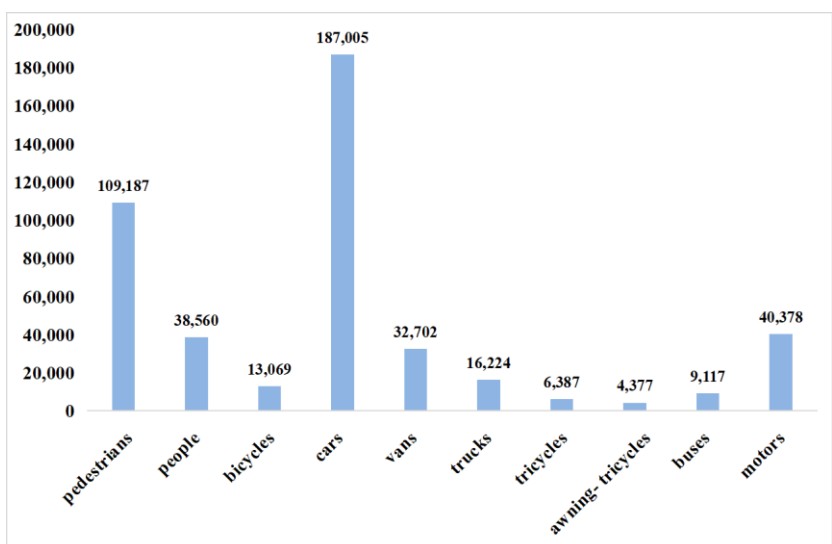

**Figure 11.** Distribution of labels within the dataset.

4.2.2. Experimental Environment

The deep learning process ran in a server with a relatively powerful GPU and the server configuration is described as follows: Intel Core I5 12 400F processor, DDR4 32 G memory, and GeForce RTX 3060 12 G GPU. Software versions were used as follows: Python version 3.7.7, Pytorch version 1.7.1, and Cuda version 10.1. The training was performed

using a stochastic gradient descent with a momentum value of 0.937, a weight decay factor of $5 \times 10^{-4}$, an initial learning rate of 0.01, a batch size setting of 4, and a period setting of 150. For geometric distortion, a random scaling and flipping method was used. The random scaling interval is (0.25, 2), and the input image is scaled randomly between 0.25 and 2, while the width and height are distorted between 0.538 and 1.857. In addition, when the flip ratio is fixed at 0.5, there is a 50% chance of flipping the image to the left or to the right. For photometric distortion, the distortion factors for hue, saturation, and luminance are set to 0.1, 0.7, and 0.4, respectively. The gamut of the images is changed after adjusting the image from RGB to HSV.

### 4.2.3. Ablation Experiment

The object detection performance of the YOLOX-IM model was evaluated using mAP50, AP-large, AP-mid, and AP-small [49]. The average accuracy (AP) for each category of mAP50 is calculated at an IoU of 0.5 and then averaged over all categories. AP-large, AP-mid, and AP-small are the AP values calculated for three different scales of objects (large, medium, and small) when the IoU is in the range of 0.5–0.95.

The formula for calculating mAP is stated as follows:

$$mAP = \frac{1}{n}\sum_{i=1}^{n} AP(i) \tag{15}$$

where $n$ is the total number of categories, $i$ is the current category, and $AP(i)$ is the AP value of the current category.

The experimental results of ablation model are shown in Table 2. Compared with the baseline network, the performance enhancement is obvious when the SAHI algorithm and coordinate correction matrix are used for data augmentation, and the detection accuracy of large, medium, and small targets is improved by 4.7%, 1.6%, and 1%, respectively. SAHI makes small objects occupy more pixels in the slices, and the data augmentation increases the diversity of training samples, which effectively improves the detection accuracy of small objects. Then, the shallow feature map $C_2$ is introduced into an FPN for feature fusion, and the small object detection accuracy is further improved by 1.3%. The results show that the feature map $C_2$ obtains rich spatial information on small targets and is suitable for small object detection in UAV aerial images. Then, ULSAM is added in the PAN stage to obtain a different attention map, highlighting the object features and weakening the background information, which improves the detection performance of the model. Finally, the loss function IoU is replaced by SIoU, which enables the bounding box to locate the target more quickly and accurately, and improves the accuracy of the network in detecting small targets.

**Table 2.** Results of ablation experiments with improved modules.

| Baseline Model (YOLOX-s) | Data Enhancement | $C_2$ | ULSAM | SIoU | Volume (MB) | mAP50 | AP-Small | AP-Mid | Ap-Large |
|---|---|---|---|---|---|---|---|---|---|
| √ | | | | | 13.85 | 36.62% | 0.103 | 0.312 | 0.422 |
| √ | √ | | | | 13.85 | 38.74% | 0.113 | 0.329 | 0.476 |
| √ | √ | √ | | | 6.7 | 41.98% | 0.124 | 0.341 | 0.495 |
| √ | √ | √ | √ | | 8.2 | 44.13% | 0.136 | 0.364 | 0.488 |
| √ | √ | √ | √ | √ | 4.55 | 44.75% | 0.142 | 0.366 | 0.506 |

Compared with the base YOLOX-s model, the improved YOLOX-IM is 67.14% less in volume, mAP50 is 8.13% more, and AP-small, AP-mid, and Ap-large are 3.9%, 5.4%, and 8.4% more, respectively. The results show that the improved model can effectively solve the problem of many small targets and complex backgrounds in UAV aerial images and achieve the lightweight requirement.

### 4.2.4. Comparison of Object Detection Algorithms

The performance of the YOLOX-IM object detection model was compared with CenterNet [50], YOLOv3 [51], D-A-FS SSD [52] and RetinaNet [53], and Faster R-CNN and Cascade R-CNN algorithms. The comparison results are shown in Table 3. The results in the table clearly show that the accuracy of the proposed YOLOX-IM model is higher than other detection models, which indicates the superior performance of the YOLOX-IM model over other detection models.

**Table 3.** Classification results of different models in the test set (mAP50).

| Model | Resolution | mAP | Pedestrian | People | Bicycle | Car | Van | Truck | Tricycle | Awning-Tricycle | Bus | Motor |
|---|---|---|---|---|---|---|---|---|---|---|---|---|
| CenterNet | - | 26.60% | 0.23 | 0.21 | 0.15 | 0.6 | 0.24 | 0.21 | 0.2 | 0.17 | 0.38 | 0.24 |
| YOLOv3 | 768 × 768 | 41.35% | - | - | - | - | - | - | - | - | - | - |
|  | 1120 × 1120 | 45.64% | 0.44 | 0.28 | 0.23 | 0.85 | 0.53 | 0.54 | 0.31 | 0.27 | 0.65 | 0.46 |
| D-A-FS SSD | - | 36.70% | - | - | - | - | - | - | - | - | - | - |
| RetinaNet | - | 35.59% | 0.27 | 0.13 | 0.14 | 0.59 | 0.50 | 0.54 | 0.25 | 0.30 | 0.59 | 0.24 |
| YOLOX-s (Baseline Model) | 640 × 640 | 36.62% | 0.31 | 0.21 | 0.15 | 0.78 | 0.41 | 0.46 | 0.22 | 0.19 | 0.58 | 0.36 |
| **YOLOX-IM (Ours)** | 640 × 640 | **47.20%** | **0.45** | **0.32** | **0.26** | **0.85** | **0.51** | **0.56** | **0.32** | **0.28** | **0.69** | **0.48** |
| FasterR-CNN | - | 33.60% | - | - | - | - | - | - | - | - | - | - |
| CascadeR-CNN | - | 43.70% | 0.43 | 0.33 | 0.21 | 0.80 | 0.49 | 0.44 | 0.32 | 0.22 | 0.62 | 0.43 |

It can be seen from Table 3 that the mAP of the YOLOX-IM model proposed in this paper is 47.20% on the test dataset, which is a 10.58% improvement over the baseline model and a 5.85% improvement over the common YOLOv3. It is worth mentioning that the YOLOX-IM model shows higher detection accuracy for the categories of pedestrians, bicycles, and motor, which also proves the effectiveness of the model improvement in this paper for small object detection under aerial images.

### 4.2.5. Comparison of Target-Tracking Algorithms

In the performance evaluation of target tracking, this study used IDSW, MOTA, and MOTP as evaluation indexes for multi-vehicle tracking results [54]. IDSW indicates the number of times the target IDs are switched during the tracking process. The multi-object tracking accuracy MOTA is an important index to measure the ability of the multi-object tracking algorithm to track vehicles continuously. The larger value denotes the higher detection accuracy, and the calculation formula is listed as follows:

$$MOTA = 1 - \frac{\sum_t (FNt + FPt + IDSW)}{\sum_t GTt} \quad (16)$$

where *FNt* is the number of vehicles not associated with the object (number of missed detections). *FPt* indicates the number of predicted trajectories of other vehicles (number of false detections). *IDSW* indicates the number of ID changes of the object.

The multi-object tracking accuracy *MOTP* represents the overlap between the tracking frame of all target vehicles and the actual frame of the target vehicle:

$$MOTP = \frac{\sum_{t,i} IOU_{t,i}}{\sum_t Gt} \quad (17)$$

where *IOU* refers to the average intersection ratio of all target tracking frames to all actual frames, and *Gt* refers to the number of successful matches in all frames.

The experiments used YOLOX-IM from Section 3 as a detector, used the Deep-Sort algorithm as a tracker, and included videos of vehicles in the VisDrone2021 dataset as tests, which include traffic scenes with different views and intersections, different weather, and different occlusion conditions. Comparisons of the commonly used target tracking algorithms were carried out. The specific comparison results are shown in Table 4.

**Table 4.** Tracking performance comparison of four multi-object tracking algorithms.

| Tracking Algorithm | MOTA | MOTP | IDSW |
|---|---|---|---|
| SORT | 60.9 | 79.5 | 164 |
| Deep-Sort | 61.2 | 81.3 | 99 |
| YOLOv3 + Deep-Sort | 65.8 | 84.4 | 71 |
| YOLOX-IM + Deep-Sort | 71.3 | 85.9 | 53 |

It can be seen that the SORT algorithm using the Faster R-CNN detector appears to have a higher number of IDSW, and the object tracking performance of the Deep-Sort algorithm, also using the Faster R-CNN detector, is 0.3% higher in MOTA and 1.8% higher in MOTP compared to SORT; the number of IDSW is also significantly reduced, which indicates that the appearance-matching strategy of the Deep-Sort algorithm enhances the correlation between vehicle detection and tracking. The tracking performance of the Deep-Sort algorithm is also further improved when the detector is replaced with the YOLOX-IM, with a 4.6% improvement in MOTA and a 3.1% improvement in MOTP, indicating that the detection algorithm can also improve the accuracy and precision of multi-object tracking.

*4.3. Control Experiments*

4.3.1. Field Experiments

This study further verified the practical application results and the accuracy of the vehicle speed extraction of YOLOX-IM-DeepSort model. Then, the experimental vehicles set up in the 12 traffic scenarios of Table 1 were detected and tracked, and only the AP values of the cars in the experimental results were taken as the basis for the object detection accuracy. In the algorithm processing, the confidence level is taken every two frames since the experimental vehicle appears, and it is worth noting that the confidence level is used to represent the AP value of the car for the convenience of describing the accuracy. The process is shown in Figure 12, and a total of 50 consecutive confidence levels were selected for comparison, with the relevant results shown in Figures 13–15.

The location of the field experiment was chosen in Tianjin, China, on Jingang Avenue, which is a four-lane two-direction urban expressway, as shown in Figures 16 and 17. The field experiment was conducted during the daytime and nighttime on 4 November 2022.

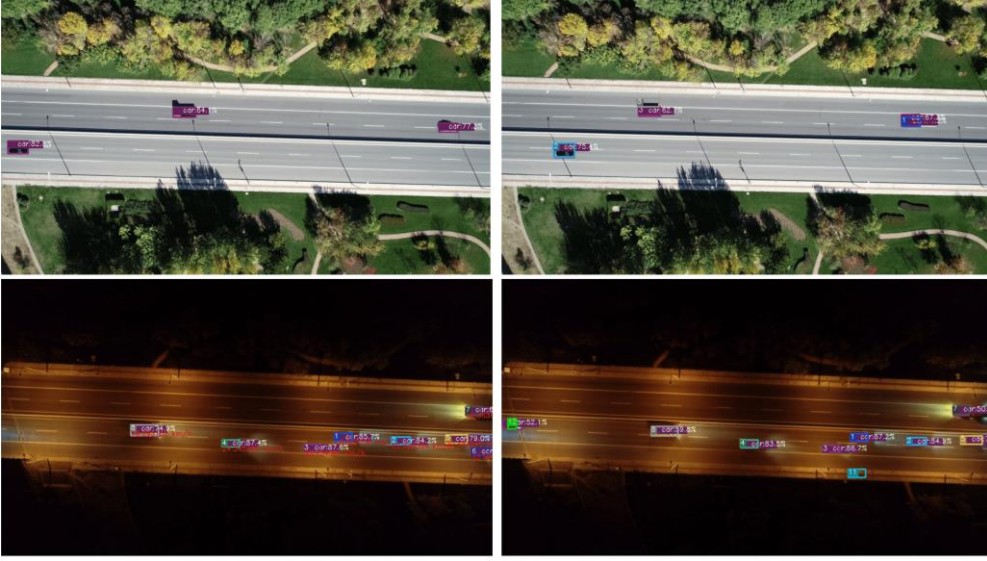

**Figure 12.** The process of taking confidence level in aerial video detection.

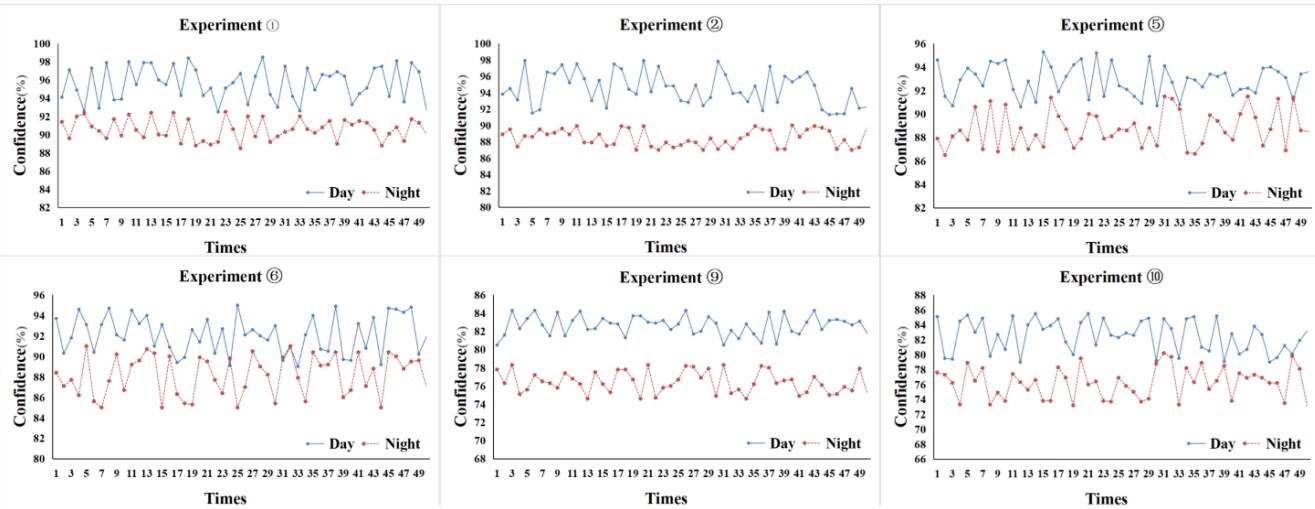

**Figure 13.** Vehicle detection accuracy (vehicle at uniform speed).

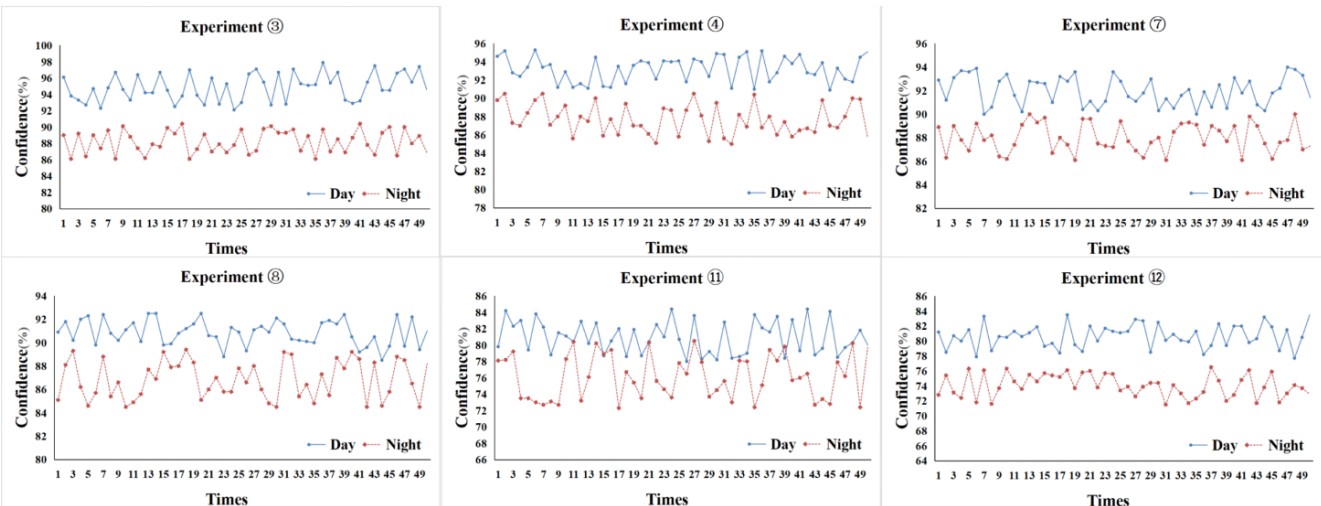

**Figure 14.** Vehicle detection accuracy (vehicle at variable speed).

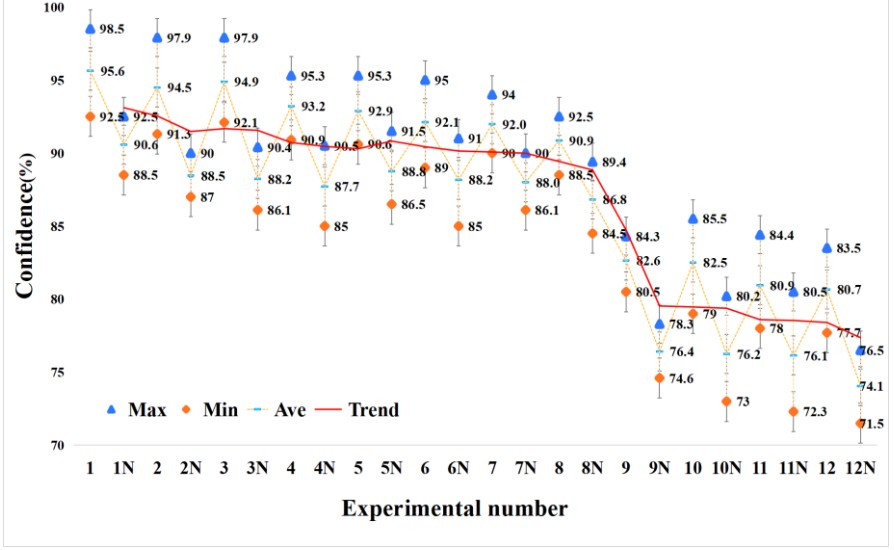

**Figure 15.** Trend graph of the change in the highest value of the confidence level.

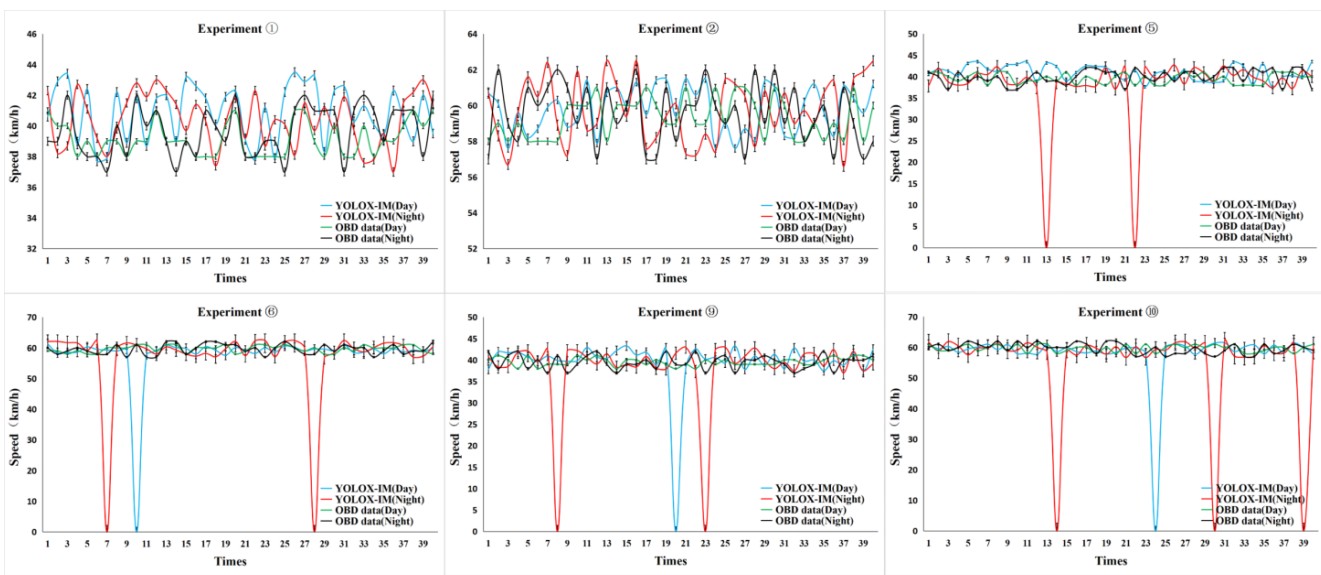

**Figure 16.** Extracted speeds vs. ground-truth speeds (vehicle at uniform speed).

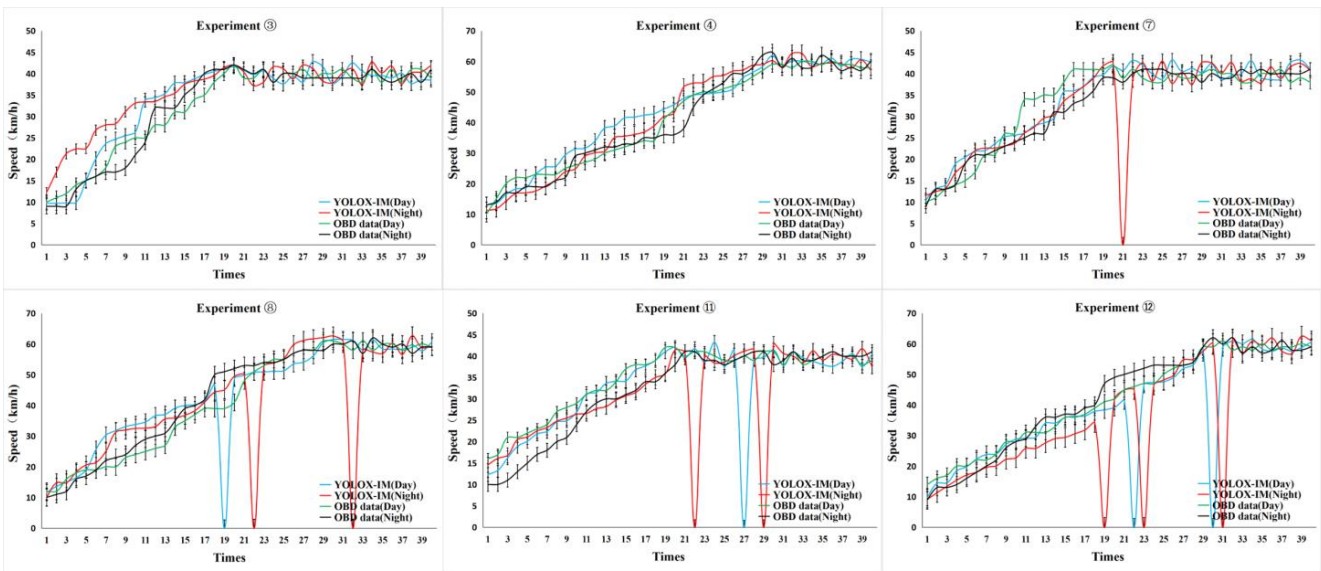

**Figure 17.** Extracted speeds vs. ground-truth speeds (vehicle at variable speed).

As shown in Figures 13 and 14, the titles of the pictures refer to the experiment numbers in Table 1. From these pictures, it can be seen that, when the height of the UAV is 30 m or 50 m, the average vehicle detection accuracy of the YOLOX-IM model is relatively high regardless of whether the speed of the experimental vehicle is uniform or variable, and the vehicle detection accuracy at low brightness is only slightly reduced compared to the vehicle detection accuracy during the daytime. However, when the UAV height is 100 m, the vehicle detection accuracy at this time is significantly reduced compared with that at other heights, and the vehicle detection accuracy at low brightness will be further significantly reduced.

From Figures 13–15, it can be seen that the object detection algorithm proposed has little impact on the accuracy of object detection when processing the video of experimental vehicles taken by UAV at a constant or variable speed, with a fluctuation range of ±1.5%; the object detection accuracy fluctuates little when the monitoring height setting of UAV changes from 30 m to 50 m and, with a fluctuation range of 2%, the purpose of traffic monitoring is still achieved with high accuracy. However, when the monitoring height of

the UAV continues to increase to 100 m, the object detection accuracy decreases more, and the average confidence level is only about 80%, which shows that a further reduction in the size of the object brings a greater challenge to the algorithm. It is worth noting that, in the nighttime aerial drone video, when the drone height is in the range of 30–50 m, the average confidence level obtained by the algorithm only decreases by 4.0–5.9% compared with the average confidence level in the daytime, regardless of whether the target vehicle is traveling at a uniform speed or at a variable speed. However, when the UAV height is 100 m, the average confidence level at night decreases more compared with the average confidence level during the day. Therefore, it is concluded that the proposed model can meet the accuracy requirements of daily traffic monitoring when the UAV is located in the height range of 30–50 m.

4.3.2. Accuracy Analysis of Vehicle Speed Extraction

Data based on GNSS and OBD acquisitions were chosen to measure the accuracy of the vehicle speed of the experimental vehicle obtained through the model. As the OBD data were recorded at a frequency of 5 Hz and the GNSS was generated at 20 Hz, the two data sources were interpolated. Additionally, similar calculations were carried out for the velocity data extracted by the model. The error comparison is shown in Figures 16 and 17.

The relative error was used to measure the accuracy of the sample data, and the goodness-of-fit index of the quoted curve regression [55] was not considered for discussion. These figures show that the largest relative error between the true ground sample value and the experimental sample value of the speed of the experimental vehicle occurred in the nighttime experiment of ⑫, with a maximum relative error of 10.43%; the smallest relative error occurred in the daytime experiment of ④, with a minimum relative error of 2.68%, while the relative errors of the other groups of experiments generally ranged from 3% to 6%. It is worth noting that, when the UAV is at 50 and 100 m, the target-tracking algorithm will incur an IDSW, resulting in a brief loss of the experimental vehicle being tracked, when the speed estimation algorithm will treat the speed of the experimental vehicle as 0. This is the reason for the abrupt curve change points in Figures 16 and 17, which occur more frequently in the nighttime experiments. In addition, this experiment found that the number of IDSWs increased more when the UAV altitude was 100 m than when the UAV altitude was 50 m. The above accuracy analysis shows that the YOLOX-IM-DeepSort model proposed is most suitable for traffic monitoring platforms where the UAV height is set at 50 m, and also for main roads or expressways where the average traffic speed is around 60 km/h.

**5. Conclusions**

In this study, a YOLOX-IM-DeepSort model was proposed for the extraction of vehicle traffic parameters in aerial images with different brightness. The dataset was processed with HSV domain data enhancement (adjustment of brightness, HSV disturbances), matrix transformation processing, and the SAHI algorithm for data enhancement, which improved the quality of the training and validation data. In the model structure, the number of detection heads for detecting small objects after fusion of $C_2$ and effective feature maps was increased, effectively improving the model's ability to detect small objects. Furthermore, ULSAM was also added to obtain different attention maps to highlight the object information and weaken the background information. Finally, the loss function of the boundary regression was optimized to make the boundary regression faster and more accurate, thus improving the model training speed and detection accuracy. The results of the ablation experiments on the VisDrone2021 dataset showed that, compared to the baseline YOLOX-s model, the improved YOLOX-IM showed a 67.14% volume reduction and an 8.13% increase in mAP50, with 3.9%, 5.4%, and 8.4% increases in AP-small, AP-mid, and Ap-large, respectively.

In addition, the VisDrone2021 dataset was adopted to verify the object detection and object tracking performance of YOLOX-IM-DeepSort, and experimental results showed that

the average vehicle detection accuracy is 85.00% and the average vehicle tracking accuracy is 71.30% at various brightness levels, both of which are better than the common model. However, it was argued that the good performance of the traffic parameter extraction model cannot be demonstrated in common datasets alone, as this would lack a metric and the model would lack interpretability. A field experiment was conducted using an experimental vehicle equipped with OBD, GNSS, and a DJI Phantom 4 RTK drone. The results showed that when the drone height ranges from 30 m to 50 m, the average accuracy of nighttime vehicle detection is only 4.0–5.9% lower compared to the daytime detection results, and the average traffic parameter extraction accuracy is 97.32% in daytime and 91.59% in nighttime. In summary, the most applicable traffic-monitoring conditions for the traffic parameter extraction model proposed in this paper are clear day and night with the UAV height between 30 and 50 m.

## 6. Discussions

The proposed traffic parameter extraction model is promising, however, there are still some limitations. For example, the increased detection head improves the detection accuracy but affects the detection efficiency of embeddable applications. Secondly, although this paper achieves the extraction of traffic parameters under low brightness, it does not take into account the situation whereby foreign objects might invade the image under extreme weather, which would affect the accuracy of vehicle detection and tracking. Furthermore, future work is addressed as follows: (1) In the field of object detection, the attention mechanism of the model is improved based on the latest YOLO model to improve feature extraction, the feature pyramid of the model is improved to fully integrate multi-scale features of the image, and the loss function of the model is optimized to improve the localization of objects. (2) It is necessary to further explore the effectiveness of the YOLOX-IM-DeepSort model in extreme weather conditions (such as heavy rain, snow, dust storms, and haze) and make targeted improvements to enable the model to be used in all weather conditions. (3) For data validation, in addition to using OBD as well as GNSS, real data from different sources (millimeter wave radar data, LiDAR point cloud data, etc.) are combined to further validate the accuracy of the traffic parameter extraction model.

**Author Contributions:** Conceptualization, J.L. and X.L.; data curation, J.L. and X.L.; funding acquisition, X.L. and Q.C.; methodology, J.L. and S.N.; validation, J.L. and X.L.; writing—original draft, J.L. and X.L.; writing—review and editing, J.L. and X.L. All authors have read and agreed to the published version of the manuscript.

**Funding:** This work was supported by the National Natural Science Foundation of China (No. 51408417) and the Science and Technology Plan Project of Tianjin, China (No. XC202028, 2022ZD016, 22YDTPJC00120).

**Institutional Review Board Statement:** Not applicable.

**Informed Consent Statement:** Not applicable.

**Data Availability Statement:** Not applicable.

**Conflicts of Interest:** The authors declare no conflict of interest.

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
