# Peer review of "A Traffic Parameter Extraction Model Using Small Vehicle Detection and Tracking in Low-Brightness Aerial Images"

_sustainability, doi:10.3390/su15118505_

Round 1

Reviewer 1 Report

Dear Authors,

Your manuscript presents an interesting approach to traffic modelling. However, I have some suggestions for improving the quality and detail as follows:

1. The literature review should be expanded, with particular emphasis on papers from the last five years.

2. Figures are illegible. The font and pictures should be bigger. Avoid languages other than English (Fig. 12).

3. How did you get the optimal hyperparameter values? Many studies have used genetic algorithms to optimize an invariant parameter in the training process? If you haven't tried to solve this problem, the neural network can be improved.

4. In the Conclusions section, directions for further research should be indicated.

Author Response

Thanks for your reviewing,Please see the attachment.

Reviewer 2 Report

The solution proposed in the paper contributes to scientific knowledge because  proposed  YOLOX-IM-DeepSort model improved the object detection performance in low-brightness conditions accurately and efficiently,  especially from dron perspective, but the paper :

1. Did not technically properly process the proposed solution:

 - The abstract has 337 words while the maximum  is 250 words and there are abbrevations which is not recommended. Also abstract should be an objective representation of the article with obligatory notated it's contribution to scince.

-Figures a posebno 7, 12 ,14, and 18-22 are not transparent and therefore not well understood.

2. The organization of the work by chapters is not adequate to the instructions for the authors of the journal, and therefore the presented content is quite incomprehensible. The number of considered references is certainly too small, out of a total of 5 in chapter 2. Realted work??? and that chapter, with such a small content, certainly had to be part of chapter 1.Intrduction according to the instructions. Chapter 3. 3. Traffic parameter extraction process and model deals with the material and methods used in the work, so it should be called that and organized by subsections. Chapters 4 and 5 should include the results and the discussion and are called that, which would free the conclusion both in the name and in the content of the unnecessary part of the discussion.

3. The subject matter is not presented in a comprehensive manner:

- The paper lacks with limitations of the proposed framework and comparation proposed framework with already existing.

4. The number  from total 31 references in the paper  is too small for the publication of the paper  in such an eminent journal as SUSTAINABILITY . Especially since only 5 of that number are in the section provided for that –  Related work were must  be described more references  that deal with  the problem which is the subject of considering in this paper. 

Author Response

(The authors gave the same response as above.)

Reviewer 3 Report

In this manuscrpt, the authors proposed  YOLOX-IM-DeepSort model for the extraction of vehicle traffic parameters in aerial images with different brightness. HSV domain data enhancement (adjustment of brightness, HSV interference), matrix transformation processing and data enhancement by SAHI algorithm were performed on the dataset to improve the quality of training and validation data. After that, extensive experiments are used to validate the feasiblity of the proposed methods. 

In my opinin, the idea could be interesting, and the manuscript can be accepted after some necessary revisions as follows

1) The contents are a little tedious, and I suggest the authors reduce the paper length to at most 20 pages.

2) Some figures are not very clear, and I suggest the authors redraw some figures such as Fig.8, Figs.13-14, Figs.18-20.

3) Some typos exist, such as "im-proved"--"improved" in Abstract; in P9, after Eq.(3), there should be a comma, and "   Where, [x0, y0]"-->"where [x0, y0]". Thus, I suggest the authors make a thorough check. 

Overally, the English is good, but some typos exist and I wish the authors can make a thorough check. 

Author Response

(The authors gave the same response as above.)

Round 2

Reviewer 1 Report

Thank you very much for your responses. All my doubts were cleared. Your manuscript is high-quality.

Reviewer 2 Report

The authors accepted all my suggestions and corrected the paper according to them, so my opinion is that it can now be published in the journal.

Reviewer 3 Report

The authors have made necessary revisions, and it can be accepted at present.